# Deviation Tolerance Performance Evaluation and Experiment of Picking End Effector for Famous Tea

Yingpeng Zhu [1], Chuanyu Wu [1,2], Junhua Tong [1,2], Jianneng Chen [1,2], Leiying He [1,2], Rongyang Wang [3] and Jiangming Jia [1,*]

1   Faculty of Mechanical Engineering & Automation, Zhejiang Sci-Tech University, Hangzhou 310018, China; 201910501018@mails.zstu.edu.cn (Y.Z.); cywu@zstu.edu.cn (C.W.); jhtong@zstu.edu.cn (J.T.); jiannengchen@zstu.edu.cn (J.C.); hlying@zstu.edu.cn (L.H.)
2   Key Laboratory of Transplanting Equipment and Technology of Zhejiang Province, Hangzhou 310018, China
3   College of Mechanical and Electrical Engineering, Huzhou Vocational and Technical College, Huzhou 313000, China; 2012005@hzvtc.net.cn
*   Correspondence: jarky@zstu.edu.cn; Tel.: +86-0571-86843741

**Abstract:** Accurately obtaining the posture and spatial position of tea buds through machine vision and other technologies is difficult due to the small size, different shapes, and complex growth environment of tea buds. Therefore, end effectors are prone to problems, such as picking omission and picking error. This study designs a picking end effector based on negative pressure guidance for famous tea. This end effector uses negative pressure to guide tea buds in a top-down manner, thereby correcting their posture and spatial position. Therefore, the designed end effector has deviation tolerance performance that can improve the picking success rate. The pre-experiment is designed, the tip of apical bud is referred to as the descent position, and the negative pressure range is determined to be 0.6 to 0.9 kPa. A deviation tolerance orthogonal experiment is designed. Experimental results show that various experimental factors are ranked in terms of the significance level of the effect on the average success rate, and the significance ranking is as follows: negative pressure ($P$) > pipe diameter ($D$) > descent speed ($V$). An evaluation method of deviation tolerance performance is presented, and the optimal experiment factor-level combination is determined as: $P$ = 0.9 kPa, $D$ = 34 mm, $V$ = 20 mm/s. Within the deviation range of a 10 mm radius, the average success rate of the negative pressure guidance of the end effector is 97.36%. The designed end effector can be applied to the intelligent picking of famous tea. This study can provide a reference for the design of similar picking end effectors for famous tea.

**Keywords:** famous tea; end effector; deviation tolerance; orthogonal experiment; evaluation method

## 1. Introduction

Tea is a natural health drink loved by people from all over the world [1,2]. Famous tea has high drinking value and economic benefits and is the pillar of the tea industry. At present, the picking of ordinary tea has been mechanized [3,4], whereas the picking of famous tea still relies on manual labor. With the increasing shortage of the tea picking labor force, the phenomenon of a "tea picker shortage" is becoming increasingly serious, thereby limiting the development of the tea industry [5,6]. Realizing the mechanized picking of famous tea is an objective requirement and inevitable trend for the sustainable development of the tea industry.

At present, research on mechanized picking of famous and excellent tea is still in the exploration stage, and the research direction mainly focuses on the identification and location of tea buds. Early research methods are mainly based on the color, shape, texture, and other characteristics of tea leaves to identify the tea buds [7–9]. Such methods have poor robustness and low accuracy. Deep learning has developed rapidly. Many researchers have begun to use network models, such as Faster R-CNN [10] and YOLO [11], to identify

the tea buds and locate them on 2D images. However, many problems remain to be solved due to the small size, different shapes, and complex growth environment of tea buds, and the robustness, accuracy, and efficiency of algorithms need to be improved.

In the actual picking conditions, end effectors are prone to problems, such as picking omission and picking error [12–14]. Thus, error compensation is required. Mehta and Burks [15] proposed a hybrid translation controller based on pursuit guidance, with control accuracy of approximately 15 mm. Wang et al. [16] designed an efficient strawberry harvesting end effector with large misalignment tolerance. This end effector can complete picking within the positioning error of ±7 mm, and the success rate of the indoor picking test was 97.7%. Ye et al. [17] proposed a dynamic positioning error analysis method to guide the fault-tolerant design of an end effector, and the success rate of the experiment was more than 90%. Zou et al. [18] designed a limited universal fruit-picking end effector, and the success rate of indoor and outdoor picking experiments was more than 84% and 78%, respectively. Xiong et al. [19] designed a novel cable-driven gripper with perception capabilities for strawberry-picking robots. The gripper is equipped with three infrared sensors to correct the position error. In the field test, the success rate of picking isolated strawberries was 96.77%. The abovementioned error compensation methods mainly include visual servo to expand the picking range of end effector and multisensor cooperative positioning. However, they cannot meet the requirements of picking tea buds in terms of the positioning accuracy and space occupation.

Different from many crops, the picking objects of famous tea are branches and leaves rather than fruits. Traditional end effectors are difficult to apply. Thus, developing new picking end effectors is necessary. Qin et al. [20] developed a picking end effector, and the intact rate of tea buds harvested was approximately 76.6%. Hao et al. [21] developed a bionic picking finger, and the picking success rate in the preliminary indoor experiment was approximately 70%. Motokura et al. [22] used a three-finger gripper on the end of a Kinova Jaco robotic arm to complete the tea picking action. Most of the existing picking end effectors for famous tea adopt some simple mechanical structures and have poor error compensation ability, which cannot ensure the success rate of picking and the intact rate of tea buds. Therefore, a deviation tolerance design of picking end effectors for famous tea is necessary.

This study aims to explore the picking technology of famous tea. The specific objectives are: (1) to study the harvesting mechanism of famous tea and develop a picking end effector with deviation tolerance performance based on negative pressure guidance; and (2) to establish a set of deviation tolerance performance evaluation methods to provide a reference for the design of similar end effectors.

## 2. Materials and Methods

### 2.1. Physical Properties of Tea

As shown in Figure 1, the picking object in this study was one-bud double-leaves, namely, the tea bud above the third leaf node. Determining the relevant physical characteristics of tea, including the overall dimension, shear force $F$, and average growth region area $S_A$, is necessary to make the end effector succeed in picking tea buds.

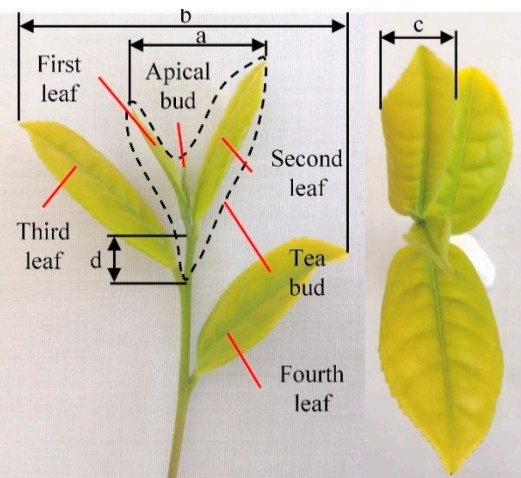

**Figure 1.** Schematic of tea characteristic parameters. Overall dimension: a—first leaf span, b—second leaf span, c—leaf width, d—node spacing.

The physical properties of tea were measured at the Lishui Comprehensive Test Station of the National Tea Industry Technology System on 24 October 2020. As shown in Figures 2 and 3, the measurement samples of the first five physical characteristics are 100 fresh tea leaves picked randomly, and the tea variety is tulip. The measuring tools are an AIRAJ brand digital caliper and a stem shear characteristic measuring instrument. As shown in Figure 4, the measuring tool of $S_A$ is a self-made square frame with a side length of 400 mm. The measurement method is to place the frame on the canopy of the tea tree, and the number of tea buds in the frame is $x$. $S_A$ can be calculated by using Formula (1), and a total of 20 measurements are made.

$$S_A = \frac{400 \times 400}{x},$$
(1)

$x$—the number of tea buds in the frame.

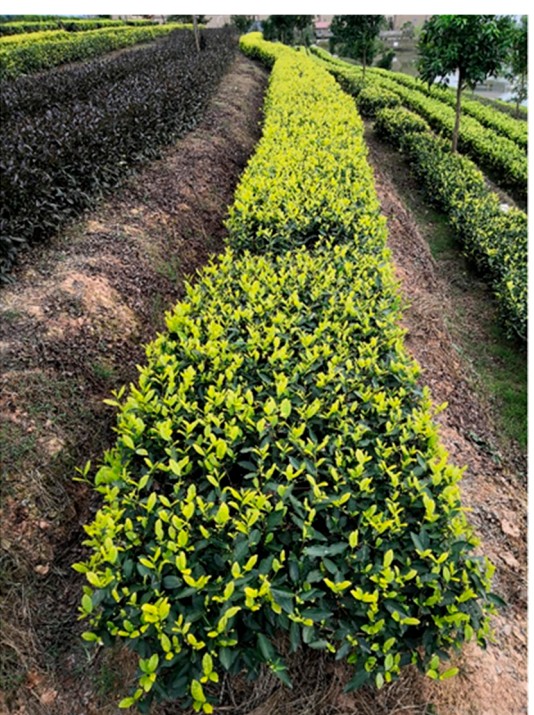

**Figure 2.** Source of measurement samples.

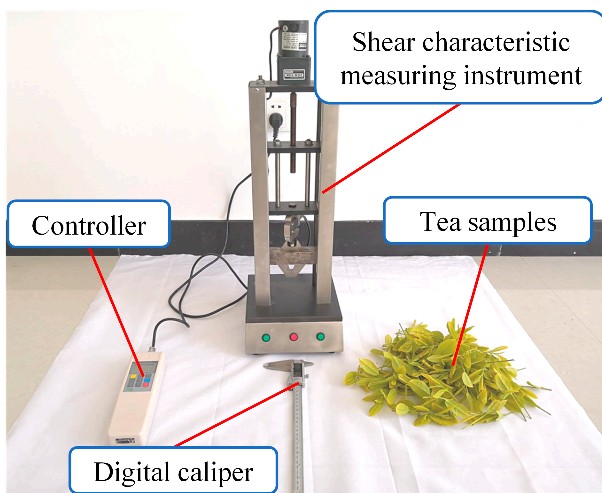

**Figure 3.** Measurement of the physical properties of tea.

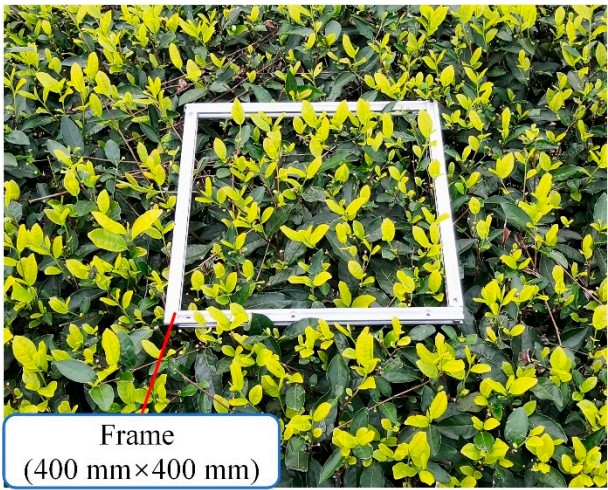

**Figure 4.** Measurement of average growth region area.

The measurement results are statistically sorted, as shown in Figure 5 and Table 1. The results show that large individual differences are found in the tea buds, thereby affecting the effect of negative pressure guidance. Therefore, the end effector is required to have strong adaptability. The growth density of tea buds is different. The average growth region area of tea buds in the densest place is 3200 mm$^2$, thereby requiring the end effector to avoid the nontarget objects.

**Table 1.** Statistical table of the physical properties of tea.

| Parameter Name | Statistical Parameters | | | |
|---|---|---|---|---|
| | Average Value | Maximum Value | Minimum Value | Standard Deviation |
| $a$ (mm) | 26.90 | 43.70 | 14.70 | 6.13 |
| $b$ (mm) | 59.78 | 89.50 | 38.80 | 10.30 |
| $c$ (mm) | 11.39 | 20.80 | 7.00 | 2.97 |
| $d$ (mm) | 13.55 | 22.80 | 7.10 | 3.17 |
| $F$ (N) | 7.64 | 10.56 | 5.15 | 1.17 |
| $S_A$ (mm$^2$) | 4017.51 | 4571.43 | 3200.00 | 360.68 |

$a$—first leaf span, $b$—second leaf span, $c$—leaf width, $d$—node spacing, $F$—shear force, $S_A$—average growth region area.

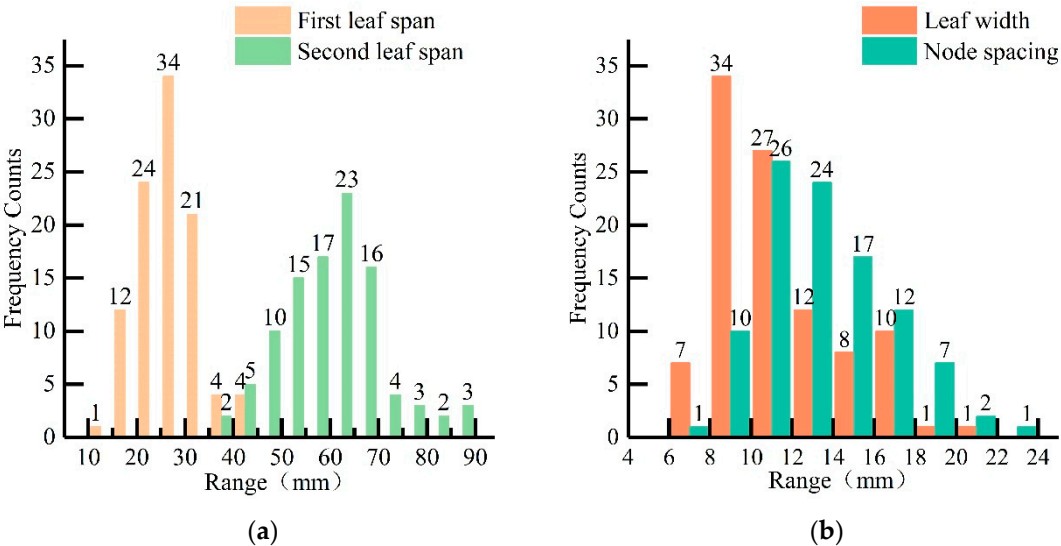

**Figure 5.** Frequency distribution of dimension parameters. (**a**) Frequency distribution of first leaf span *a* and second leaf span *b*, (**b**) Frequency distribution of leaf width *c* and node spacing *d*.

### 2.2. End Effector Structure and Principle

In accordance with the growth characteristics of tea leaves, the tea buds are mainly scattered on the canopy surface. Thus, the top-down picking method can reduce interference. A negative pressure guidance method was adopted in this study to solve the positioning and random errors. Its principle is to attract the tea bud with the airflow generated by the negative pressure. This condition can effectively improve the picking success rate of the end effector.

The end effector was designed in combination with lightweight design requirements, as shown in Figure 6, including the picking pipe, rope drive mechanism, and shearing mechanism. As shown in Figure 7, the picking pipe was formed by 3D printing, and the upper pipe opening was connected to a vacuum suction machine. The flange was fixed on the moving platform of the parallel manipulator, one end of the wire rope pipe of the rope drive mechanism was fixed on the flange, and the shear mechanism was fixed on the boss. As shown in Figure 8, the rope drive mechanism included a mounting support, steering gear, Arduino Mega 2560, wire rope pipe, and wire rope. Its function is to drive the shear mechanism for completing the shearing work. Its advantage is that the driving source is set on the frame, thereby reducing the burden of the parallel manipulator and avoiding limiting the movement speed of the manipulator. As shown in Figure 9, the shearing mechanism adopted a gear transmission, and two specially shaped blades were driven by the wire rope to complete the shearing. Its advantages are simple driving, small volume, and the blade being located directly below the lower pipe opening. Thus, interfering in the initial and shear states is difficult.

During operation, the end effector first reaches above the specified shear point to avoid interference. At this time, the vacuum suction machine begins to work, and negative pressure is generated at the lower pipe opening of the picking pipe. The parallel manipulator is driven to descend vertically so that the end effector reaches the shear point position. In this process, the tea bud is guided into the picking pipe with the airflow generated by negative pressure. The steering gear is controlled to rotate, and the wire rope is driven to tighten so that the shearing mechanism can complete the shearing. The tea buds are collected by negative pressure. At the same time, the steering gear is controlled to reset, and the shear mechanism returns to the initial position under the action of the spring.

The working principle of the end effector was analyzed. When the negative pressure guidance is successful, the probability of shear failure is minimal. Therefore, the success rate of picking depends mainly on the success rate of the negative pressure guidance.

Without considering the positioning error, the negative pressure of the vacuum suction machine, the pipe diameter of the lower pipe opening, and the descent speed of the picking pipe are the influencing factors of the negative pressure guidance.

The tea buds with a first leaf span less than 35 mm accounted for 92% of the samples, in accordance with the statistical results. Considering that negative pressure guidance can reduce the leaf span, the diameter of the lower pipe opening can be slightly smaller than the first leaf span. Thus, the minimum diameter of the lower pipe opening is 30 mm. The minimum value of the second leaf span was 38.8 mm. The maximum diameter of the lower pipe opening was set to 38 mm to reduce the inhalation of nontarget objects, such as the third and fourth leaves. Therefore, the pipe diameter range of the lower pipe opening was determined as 30–38 mm, and the three levels were set as 30, 34, and 38 mm, respectively.

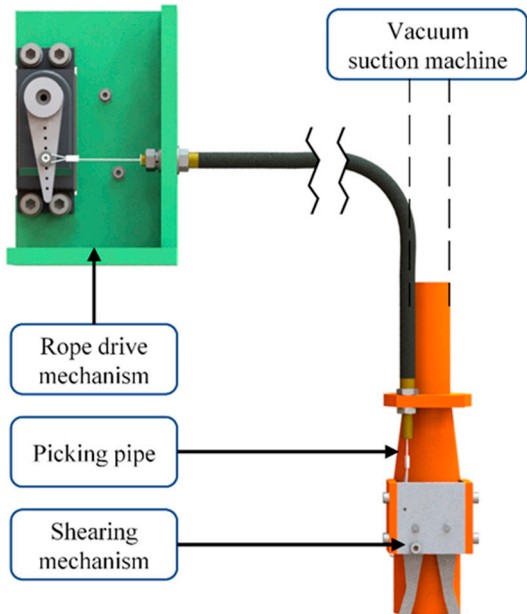

**Figure 6.** End effector.

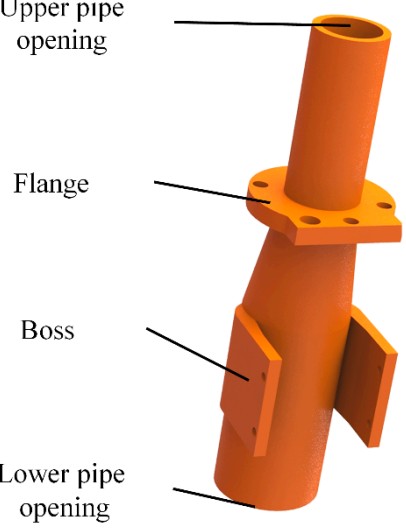

**Figure 7.** Picking pipe.

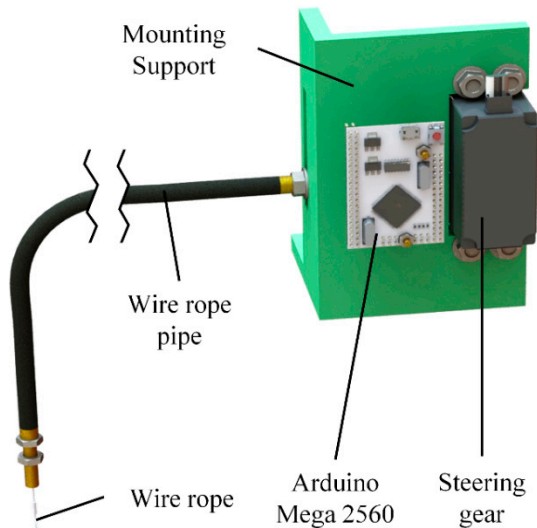

**Figure 8.** Rope drive mechanism.

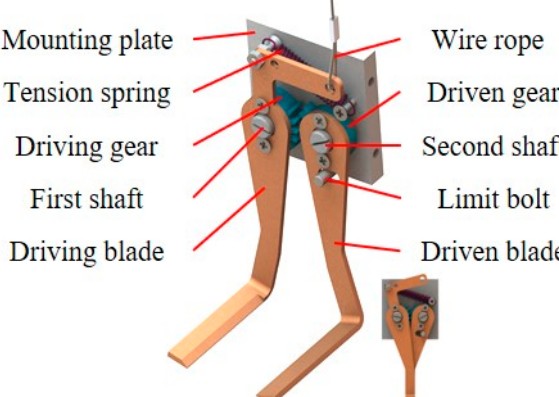

**Figure 9.** Shearing mechanism.

*2.3. Design of Experiment*

2.3.1. Experimental Conditions

The experimental platform was set up, as shown in Figure 10. The experimental equipment included the vacuum suction machine, parallel manipulator, digital vacuum gauge, laptop, high-speed camera, and light source. The high-speed camera used in this study was the Phantom VEO 340L (York Technologies Co., Ltd., Hong Kong, China). When the resolution is 2560 × 1600, the FPS is 800. The vacuum suction machine utilized was the 6281D model (Shanghai Yili Electric Co., Ltd., Shanghai, China). After modification, the maximum internal negative pressure is 0.9 kPa, which can be adjusted in a stepless manner with a knob. The brand of the digital vacuum gauge is Japanese Sanliang and the model is DP360. Its measuring range is ±10 kPa and the accuracy is 0.3%.

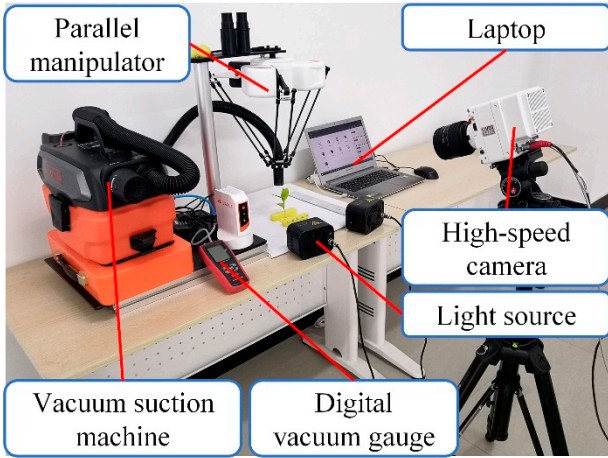

**Figure 10.** Experimental platform.

As shown in Figure 11, a ruler, a guide pipe with pipe diameters of 30, 34, and 38 mm (referred to as D30, D34, and D38, respectively), a fixed seat, end cap 1, end cap 2, and a locating pin were required. The fixed seat was used to fix the tea leaves. As shown in Figure 12, end cap 1 and the locating pin constitute location component 1, which can be used to locate the center position of D30 and D38. End cap 2 and the locating pin constitute location component 2, which can be used to locate the center position of D34.

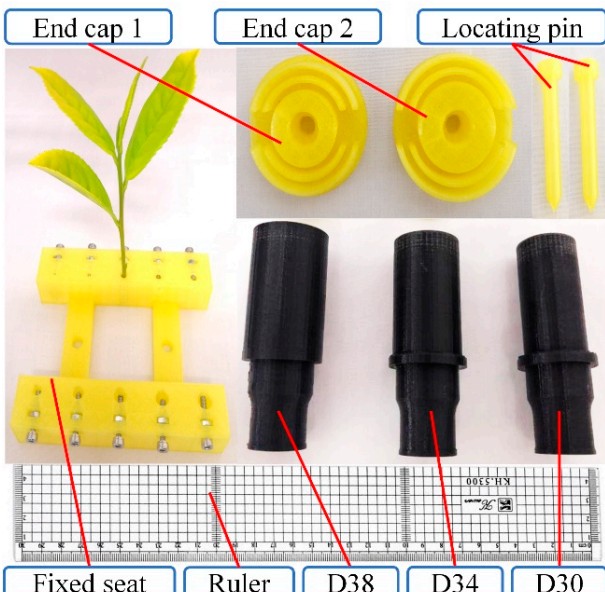

**Figure 11.** Experimental supplies.

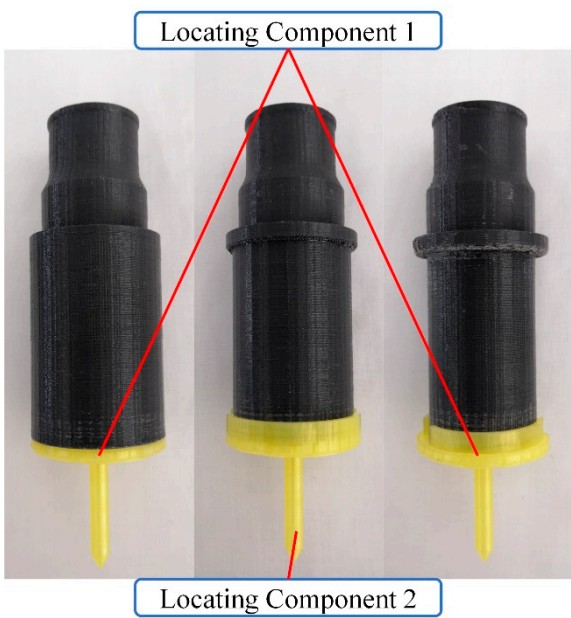

**Figure 12.** Assembly diagram of locating components.

### 2.3.2. Pre-Experiment

Multiple groups of experiments should be conducted to avoid losing generality due to the large individual differences of the tea buds. At the same time, uniformly regulating the placement of tea leaves is necessary to avoid the influence of placement errors on the test results. As shown in Figure 13, O-XYZ was the coordinate system of the parallel manipulator. The tea placement principle is that the second leaf is located in the negative direction of the X-axis, and the first leaf is located in the positive direction of the X-axis. The line between the tip of the first leaf $G_1$ and the tip of the second leaf $G_2$ is parallel to the XOZ plane.

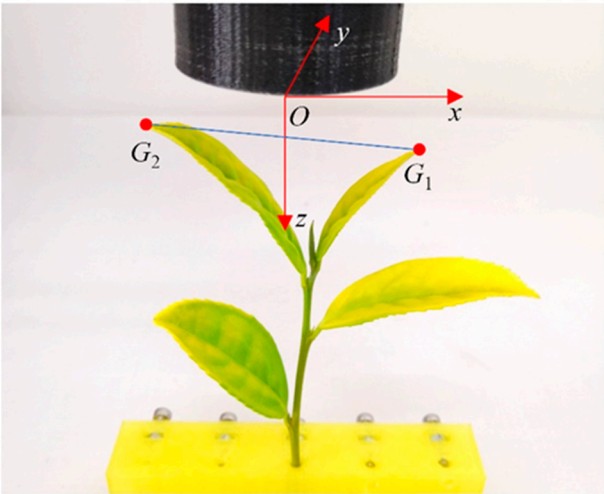

**Figure 13.** Diagram of tea placement.

The descent position of the guide pipe affects the negative pressure guidance. The first step is to determine the optimal descent position through the pre-experiment and unify it in the subsequent experiment for obtaining a better experiment effect. Considering the reliability of visual positioning and negative pressure guidance, the optimal descent position should be located at the center of the first leaf span or the tip of the apical bud. In other words, when tea leaves are placed, the tip of the locating pin should be located at the

center of the first leaf span or the tip of the apical bud. The center of the first leaf span can be determined with a ruler. The test steps are as follows:

1. Randomly picked tea leaves are placed in accordance with the placement principle of tea leaves. The locating pin is used to align the center of the guide pipe with the center of the first leaf span.
2. Turn on the vacuum suction machine, control the parallel manipulator to descent, guide the tea buds with negative pressure, and use the high-speed camera to record the guiding process.
3. After reposition, use the locating pin to align the center of the guide pipe with the tip of the apical bud, and repeat step 2.

The negative pressure guidance process of two descent positions taken by the high-speed camera is shown in Figures 14 and 15, and the image evaluation system used was the Phantom Camera Control Application version 3.1. Whether the descent position is the center of the first leaf span or the tip of the apical bud when the second leaf is guided into the guide pipe, the first leaf moves some distances to the positive direction of the *X*-axis. As shown in Figure 14d, when the descent position is the center of the first leaf span, the first leaf is far from the guide pipe center. Thus, escaping from the range of negative pressure guidance is easier than when the descent position is the tip of the apical bud, resulting in the failure of guidance. Therefore, the tip of the apical bud was chosen as the descent position, and the descent position was unified as the tip of the apical bud in the follow-up experiment.

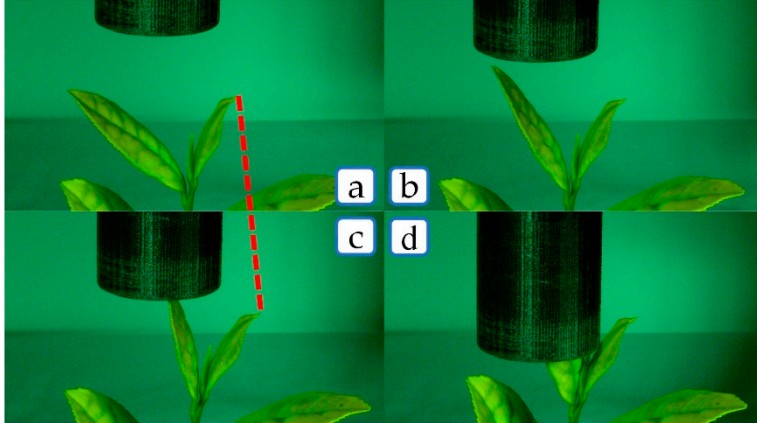

**Figure 14.** Descent position is the center of the first leaf span. (**a**) initial state, (**b**) Guidance start, (**c**) During guidance, (**d**) End of guidance.

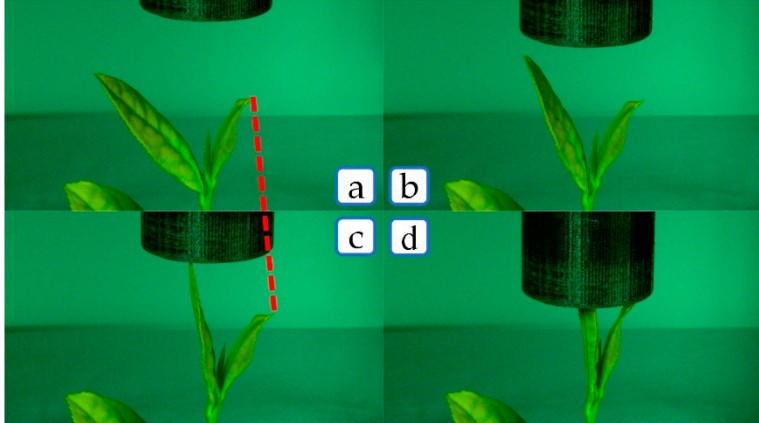

**Figure 15.** Descent position is the tip of the apical bud. (**a**) initial state, (**b**) Guidance start, (**c**) During guidance, (**d**) End of guidance.

The size of negative pressure directly affects the effect of negative pressure guidance. Determining the range of negative pressure through a pre-experiment is necessary to ensure the reliability of a deviation tolerance orthogonal experiment. The experiment samples were 60 fresh tea leaves (one-bud four-leaves) randomly picked from the tea garden, and the variety was tulip. The maximum internal negative pressure of the vacuum tea suction machine used in this study was 0.9 kPa for a long time and could be increased to 1.1 kPa for a short time. When the negative pressure exceeded 1.0 kPa, some tea leaves shook rapidly during the negative pressure guidance, thereby damaging the tea buds. This condition is attributed to turbulence formed by the fast-flowing air at the lower pipe opening. The damage rates of tea buds were 5% and 20% when the negative pressures were 1.0 and 1.1 kPa, respectively. Therefore, the maximum negative pressure was set to 0.9 kPa by considering the quality of tea bud picking and the stability of the equipment operation. The function of negative pressure guidance is to guide the tea buds into the guide pipe. The designed experiment steps to determine the minimum negative pressure are as follows:

1. The 60 randomly picked fresh tea leaves are divided into groups A, B, and C, which are used as experimental samples of D30, D34, and D38, respectively. D30 is then installed on the parallel manipulator.
2. Take the experiment samples of group A, and place the tea leaves in accordance with the tea placement principle. Set the descent speed of the parallel manipulator to 20 mm/s (low speed is good for negative pressure guidance). Turn on the vacuum suction machine, set the negative pressure to 0.2 kPa (the guiding effect is extremely weak when negative pressure <0.2 kPa), and then control the parallel manipulator to descent.
3. If the guidance is successful, the negative pressure is recorded to be 0.2 kPa. If it fails, the negative pressure increases by 0.05 kPa each time until it is successful. The negative pressure is recorded to be $(0.2 + 0.05 \, q)$ kPa, and $q$ is the times of negative pressure increase.
4. Replace the tea leaves, then repeat steps 2 and 3, and complete 20 data records.
5. Replace the guide pipe and corresponding experimental sample, and then repeat steps 2, 3, and 4.

Figure 16 shows the frequency distribution of the experiment results of the three groups. All the negative pressure values in the D30 experiment group were less than or equal to 0.6 kPa, and 95% of the negative pressure values in the D34 and D38 experiment groups were less than or equal to 0.6 kPa. Therefore, 0.6 kPa was selected as the minimum negative pressure, and the negative pressure range was 0.6–0.9 kPa.

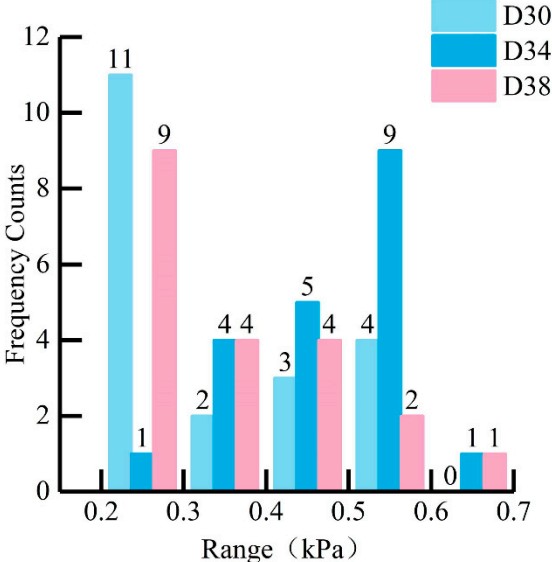

**Figure 16.** Frequency distribution of minimum negative pressure.

### 2.3.3. Design of Deviation Tolerance Orthogonal Experiment

The purpose of a deviation tolerance orthogonal experiment is to evaluate the deviation tolerance performance of the end effector under different design parameters, that is, the ability to guide the tea buds successfully when a deviation occurs in the descent position of the guide pipe. The deviation tolerance orthogonal experiment was conducted at the Lishui Comprehensive Test Station of the National Tea Industry Technology System from 2 November to 5 November 2020. The experiment samples were 180 fresh tea leaves (one-bud four-leaves) randomly picked from the tea garden, and the variety was tulip. Only 5 fresh tea leaves were picked each time and collected again after the experiment to avoid the water loss of fresh tea leaves, leading to great changes in physical properties and affecting the experimental results. This process was performed due to the large amount and time consumption of this experiment.

The deviation tolerance orthogonal experiment was conducted by using an L9(3$^4$) orthogonal table array and performing 20 tests for each combination of test conditions. The negative pressure range was 0.6–0.9 kPa, and the pipe diameter range was 30–38 mm. The speed range of the parallel manipulator was 0–100 mm/s. Considering the picking efficiency, the descent speed range was limited to 20–100 mm/s, and the factor level is shown in Table 2. The descent position was arranged in accordance with Figure 17 in the experiment, where origin $O$ is the tip of the apical bud, the evenly distributed blue points are the descent position, and the distance between adjacent points is 4 mm. If the descent position of the guide pipe shifts to the point (4,4), then the lower pipe opening of D30, D34, and D38 shifts to the position shown in the figure, which is unconducive to negative pressure guidance. In the same direction, the farther from origin $O$, the higher the failure probability of negative pressure guidance. The descent position that can be successfully guided during each experiment is marked. The success rate of guidance at different descent positions can be obtained for each group of experiments. This rate can be used to analyze and evaluate the deviation tolerance performance of the end effector.

**Table 2.** Factor level of deviation tolerance orthogonal experiment.

| Factors Level | Experimental Factors | | |
|:---:|:---:|:---:|:---:|
| | Negative Pressure $P$/(kPa) | Pipe Diameter $D$/(mm) | Descent Speed $V$/(mm/s) |
| 1 | 0.60 | 30 | 20 |
| 2 | 0.75 | 34 | 60 |
| 3 | 0.90 | 38 | 100 |

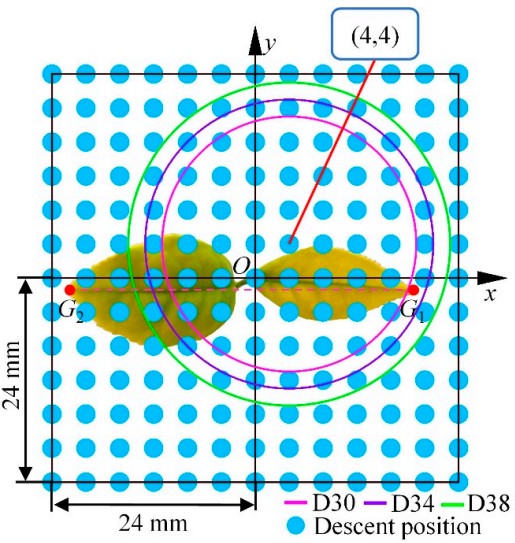

**Figure 17.** Layout diagram of descent position.

This study presents a deviation tolerance performance evaluation method that uses contour maps to statistically analyze the test results. The evaluation indicators included maximal guide area $S_M$ and average success rate $R_S$. On this basis, the level combination with less energy and time consumptions can be taken as the optimal choice. As shown in Figure 18, the range with the marginal guidance success rate of zero is defined as the maximal guidance range, and the area is $S_M$. The set of end effector deviation positions is defined as the deviation range, the deviation range center is located at the origin, the radius is the maximum positioning deviation, and the area is $S_D$. Four contour areas were assumed to intersect with the deviation range in the contour map, which are $A_1$, $A_2$, $A_3$, and $A_4$ from the outside to the inside, and the corresponding guidance success rates are $E_1$, $E_2$, $E_3$, and $E_4$. The overlapping areas of each contour area and deviation range are $B_1$, $B_2$, $B_3$, and $B_4$, and the corresponding areas are $S_1$, $S_2$, $S_3$, and $S_4$. The average success rate $R_S$ can be calculated using Formula (2).

$$R_s = (E_1\,S_1 + E_2\,S_2 + E_3\,S_3 + E_4\,S_4)/S_D, \tag{2}$$

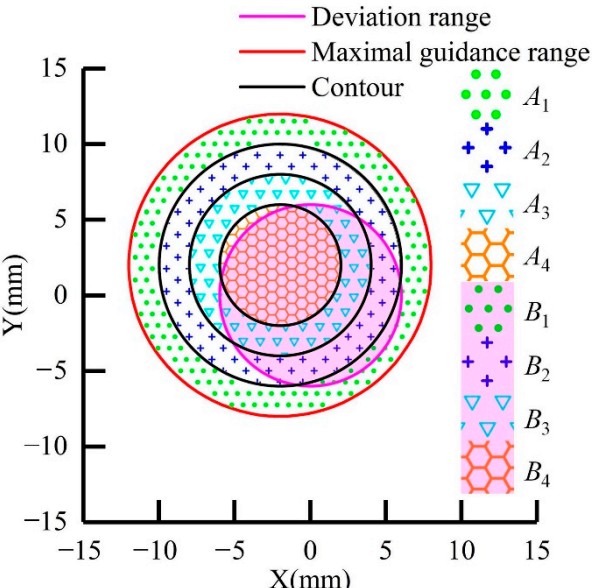

**Figure 18.** Schematic of evaluation method. $A_1$, $A_2$, $A_3$, and $A_4$—contour area, $B_1$, $B_2$, $B_3$, and $B_4$—overlapping areas of each contour area ($A_1$, $A_2$, $A_3$, and $A_4$) and deviation range.

Similarly, when the number of contour areas intersecting with the deviation range is *n*, the average success rate $R_S$ can be calculated using Formula (3).

$$R_s = \frac{\sum\limits_{j=1}^{n} E_j S_j}{S_D}, \tag{3}$$

In field picking conditions, the machine vision positioning system used in this study had a maximum positioning deviation of 10 mm in the XY direction. Thus, $S_D$ is equal to 314.16 mm$^2$. In accordance with the definition, the larger the $S_M$ is, the larger the influence range of negative pressure, and the easier it is to attract nontarget objects. The larger the $R_S$ is, the higher the average success rate of negative pressure guidance within the deviation range. $S_M$ should be determined and limited in accordance with the average growth region area of tea buds to ensure the quality and efficiency of picking. When $S_M$ satisfies the condition, the level combination of experimental factors with large $R_S$ is the best.

## 3. Results and Discussion

Figure 19 shows the deviation tolerance effect of nine experimental groups based on the contour map, verifying the influence of different design parameters on the deviation tolerance performance of the end effector. The range with the marginal guidance success rate of 95% is defined as the optimal guidance range. As clearly shown in Figure 19, when $P$ = 0.9 kPa, $D$ = 34 mm, and $V$ = 20 mm/s, the area of the optimal guidance range of the end effector is the largest. When $P$ = 0.6 kPa, $D$ = 38 mm, and $V$ = 100 mm/s, the area of the optimal guidance range of the end effector is the smallest.

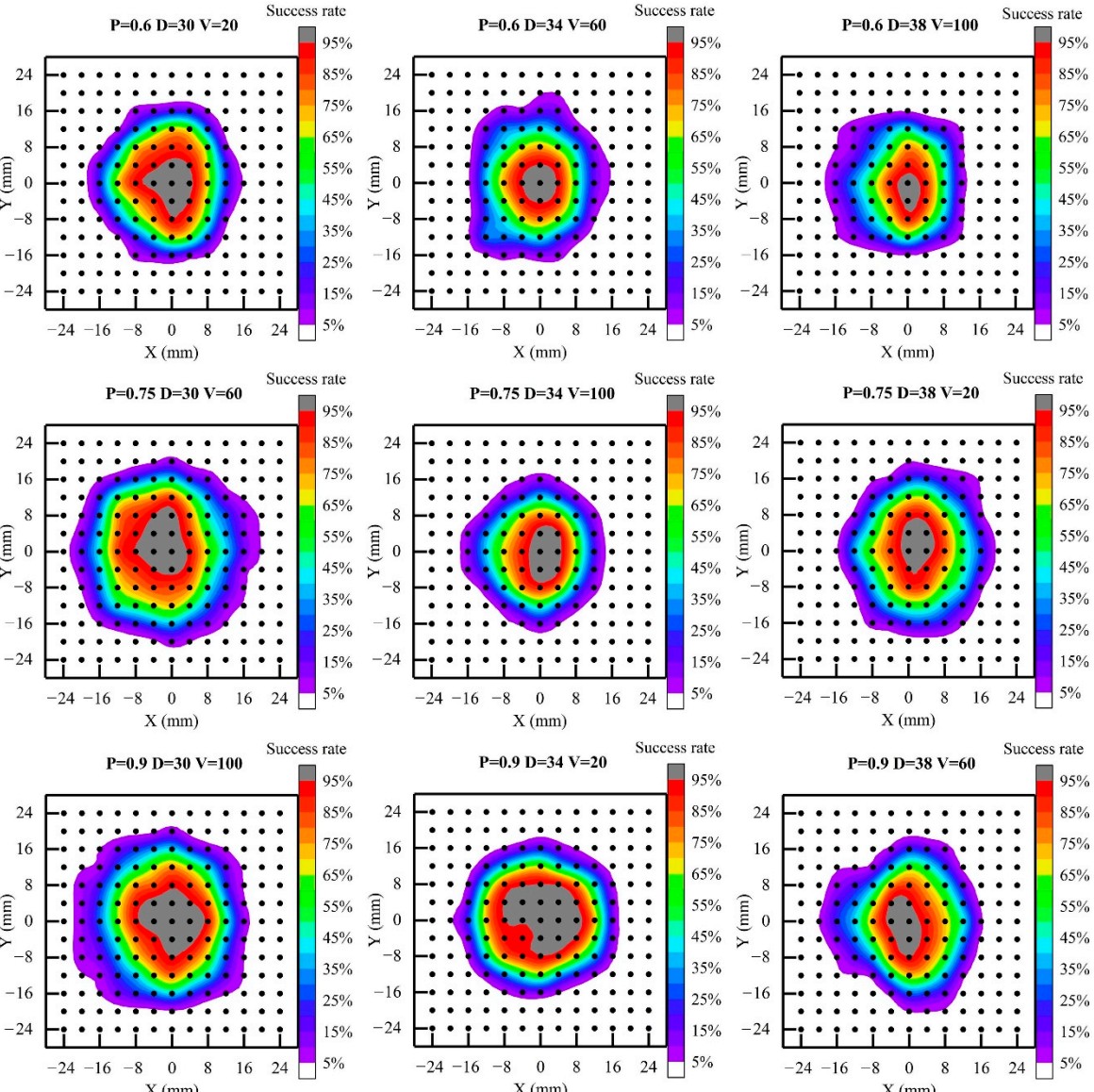

**Figure 19.** Diagram of deviation tolerance effect.

The results of the deviation tolerance orthogonal experiment are shown in Table 3, where $k_i$ represents the average value of $S_M$ when the experimental factor is at the *i* level, and $t_i$ represents the average value of $R_S$ when the experimental factor is at the *i* level, where *i* = 1, 2, 3. Range $R_1$ represents the difference between the maximum and minimum values in $k_1$, $k_2$, and $k_3$, and range $R_2$ represents the difference between maximum and minimum values in $t_1$, $t_2$, and $t_3$. Their formulas are expressed as follows:

$$R_1 = Max\{k_1, k_2, k_3\} - Min\{k_1, k_2, k_3\}, \tag{4}$$

$$R_2 = Max\{t_1, t_2, t_3\} - Min\{t_1, t_2, t_3\}, \tag{5}$$

**Table 3.** Results of deviation tolerance orthogonal experiment.

| Experimental Number | Experimental Factors | | | Evaluating Indicator | |
|:---:|:---:|:---:|:---:|:---:|:---:|
| | $P$ | $D$ | $V$ | $S_M$ | $R_S$ |
| 1 | 0.60 | 0.30 | 20 | 945.29 | 88.84% |
| 2 | 0.60 | 0.34 | 60 | 930.24 | 79.31% |
| 3 | 0.60 | 0.38 | 100 | 859.96 | 73.96% |
| 4 | 0.75 | 0.30 | 60 | 1345.64 | 89.72% |
| 5 | 0.75 | 0.34 | 100 | 832.16 | 82.64% |
| 6 | 0.75 | 0.38 | 20 | 1055.53 | 84.39% |
| 7 | 0.90 | 0.30 | 100 | 1316.43 | 94.15% |
| 8 | 0.90 | 0.34 | 20 | 1023.58 | 97.36% |
| 9 | 0.90 | 0.38 | 60 | 1018.39 | 85.32% |
| $k_1$ | 911.83 | 1202.45 | 1008.13 | | |
| $k_2$ | 1077.78 | 928.66 | 1098.09 | | |
| $k_3$ | 1119.47 | 977.96 | 1002.85 | | |
| $t_1$ | 80.70% | 90.90% | 90.20% | | |
| $t_2$ | 85.58% | 86.44% | 84.78% | | |
| $t_3$ | 92.28% | 81.22% | 83.58% | | |
| $R_1$ | 207.64 | 273.79 | 95.24 | | |
| $R_2$ | 11.57% | 9.68% | 6.61% | | |

$P$—negative pressure, $D$—pipe diameter, $V$—descent speed, $S_M$—maximal guidance area, $R_S$—average success rate, $k_i$—average value of $S_M$ when the experimental factor is at the $i$ level, $t_i$—average value of $R_S$ when the experimental factor is at the $i$ level, $R_1$—difference between the maximum and minimum values in $k_1$, $k_2$, and $k_3$, $R_2$—difference between the maximum and minimum values in $t_1$, $t_2$, and $t_3$.

The range $R_1$ values of the three experimental factors are 207.64, 273.79, and 95.24, respectively. Therefore, the order of primary and secondary factors affecting $S_M$ is pipe diameter, negative pressure, and descent speed. As shown in column $D$ of Table 3, $k_2 < k_3 < k_1$ indicates that $S_M$ decreases first and then increases as the pipe diameter increases in most of the cases observed. The main reason is that the air velocity at the lower pipe opening decreases with the increase in pipe diameter. This condition leads to the weakening of the guiding ability on the tea bud and reduces the maximal guidance range. However, the coverage range of the pipe increases with the further increase in pipe diameter, so that the maximal guidance range increases. As shown in column $P$ of Table 3, $k_1 < k_2 < k_3$ indicates that $S_M$ increases as the negative pressure increases in most of the cases observed. The main reason is that the air velocity at the lower pipe opening increases with the increase in negative pressure. This condition enhances the guiding ability on the tea bud and attracts further tea buds, thereby increasing the maximal guidance range. As shown in column $V$ of Table 3, $k_3 < k_1 < k_2$ indicates that $S_M$ increases first and then decreases as the descent speed increases in most of the cases observed. The main reason is that when the descent speed is slow, the guidance time is sufficient, and the first leaf is prone to a large offset, leading to the failure of guidance. With the increase in descent speed, the guiding time decreases, and the first leaf of the tea bud is inhaled before it moves out of the guidance range, slightly increasing the maximal guidance range. However, when the descent speed continues to increase, the second leaf of the tea bud cannot deflect in time, probably leading to the failure of the guidance and slightly reducing the maximal guidance range.

The range $R_2$ values of the three experimental factors are 11.57%, 9.68%, and 6.61%, respectively. Therefore, the order of primary and secondary factors affecting $R_S$ is negative pressure, pipe diameter, and descent speed. As shown in column $P$ of Table 3, $t_1 < t_2 < t_3$ indicates that $R_S$ increases as the negative pressure increases in most of the cases observed. As shown in column $D$ of Table 3, $t_3 < t_2 < t_1$ indicates that $R_S$ decreases as the pipe diameter increases in most of the cases observed. As shown in column $V$ of Table 3, $t_3 < t_2 < t_1$ indicates that $R_S$ decreases as the descent speed increases in most of the cases observed.

The main reason is that increasing the negative pressure or decreasing the pipe diameter can increase the air velocity at the lower pipe opening. This condition can effectively improve the guidance success rate of the tea buds near the lower pipe opening, thereby increasing $R_S$. Considering that guiding the tea buds requires time, reducing the descent speed can allow the tea buds near the lower pipe opening to have sufficient time to be guided, without excessive deviation. This condition can improve the guidance success rate of the tea buds near the lower pipe opening, thereby increasing $R_S$.

The optimal level combination of the end effector should be determined in accordance with different tea picking conditions. In accordance with Section 2.1, the average growth region area is 3200 mm² in the place where the tea buds grow most densely. The boundary is divided by a square, and the side length is 56.57 mm. As shown in Figure 20, the distance of $O_1O_2$ is 10 mm. The maximal guidance range should be controlled within the boundary to reduce the effect of negative pressure on nontarget objects. The maximal guidance range is approximated as a circle, and the maximal radius of the maximal guidance range in the figure is up to 18.285 mm, that is, $S_M$ needs to meet the following formula.

$$S_M \leq \pi \times 18.285^2 = 1050.36 \text{mm}^2 \tag{6}$$

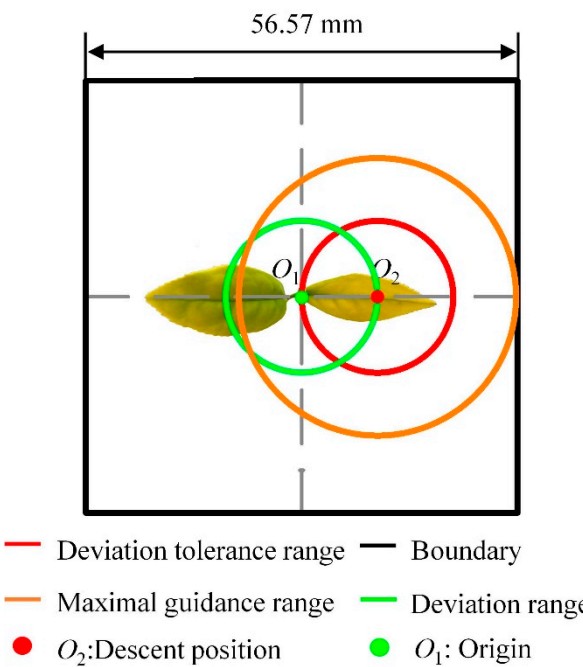

**Figure 20.** Limitation of maximal guidance range.

If the above conditions are satisfied, the level combined with the highest average success rate is optimal. Therefore, the optimal experimental level combination is $P_3D_2V_1$, which is the eighth group. At this time, the maximal guidance area $S_M$ is 1023.58 mm², and the average success rate $R_S$ is 97.36%.

The level combinations that may be better than the eighth group in terms of $R_S$ are $P_3D_1V_1$ and $P_3D_1V_2$. However, the $S_M$ of the two combinations should be greater than 1050.36 mm², which does not meet the design requirements. The verification experiment of level combination $P_3D_2V_1$ was conducted to verify the stability of the orthogonal experiment. The results are consistent with the orthogonal experiment. $S_M$ is 1009.51 mm², $R_S$ is 97.04%, and the error rates of the two indicators are 1.37% and 0.33%, respectively. The reason for the error should be the different samples of tea buds. The verification test shows that the results of the orthogonal experiment are stable and reliable.

## 4. Conclusions

In this study, a picking end effector based on negative pressure guidance was developed to solve the visual positioning error of tea buds. On this basis, a deviation tolerance performance evaluation method was presented to optimize the design parameters of the end effector. The deviation tolerance performance of the end effector was verified through experiments. The conclusions are as follows:

1.  The developed picking end effector adopts a lightweight design to meet the requirements of the rapid movement of parallel manipulators and is suitable for intensive picking operations. It also has the advantages of being simple, reliable, and low cost, which is conducive to popularization and application.
2.  The presented deviation tolerance performance evaluation method can optimize the design of the corresponding end effector for different tea picking conditions. This method can effectively improve the picking success rate of the tea buds and has guiding importance.
3.  The picking end effector based on negative pressure guidance has deviation tolerance performance and can pick the tea buds in the case of a positioning deviation in the XY direction. Under the optimal level combination condition, the area of the maximal guidance range is 1023.58 mm$^2$, and the average success rate of negative pressure guidance within the deviation range with a 10 mm radius is 97.36%. These findings can effectively improve the success rate of picking tea buds, reduce the visual positioning requirements, and accelerate the development process of the tea industry.

**Author Contributions:** Conceptualization, C.W. and J.C.; data curation, J.T., R.W., and J.J.; writing—original draft, Y.Z. and L.H. All authors have read and agreed to the published version of the manuscript.

**Funding:** This work was funded by the Post Scientist of the Modern Agriculture Industry System (Intelligent Tea Picking) of the Ministry of Agriculture and Rural Affairs of the People's Republic of China (CARS-19) and Zhejiang Provincial Natural Science Foundation of the People's Republic of China (Grant No. LY18E050025).

**Institutional Review Board Statement:** Not applicable.

**Informed Consent Statement:** Not applicable.

**Data Availability Statement:** The data presented in this study are available on request from the corresponding author.

**Acknowledgments:** We would also like to thank the samples provided by the Lishui Comprehensive Test Station of The National Tea Industry Technology System.

**Conflicts of Interest:** The authors declare no conflict of interest.

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
