# Peer review of "Deviation Tolerance Performance Evaluation and Experiment of Picking End Effector for Famous Tea"

_agriculture, doi:10.3390/agriculture11020128_

Round 1
Reviewer 1 Report
The article presented for evaluation is an interesting extension of the possibilities of automatization of tea harvesting. A large amount of technical details may not be fully consistent with the journal's profile, but I believe that it should be of interested for readers.
However, I have a few comments about work.
In formula 1, please complete the description of x.
I think it would be worth considering changing the Arduino 2560 module to eg Adrudino Due due to its higher speed, memory and general properties. This can be helpful in streamlining the software and improving the efficiency of the entire device, and easier application of an extensive recognition algorithm.
It would be useful to include additional information about the parameters of the high-speed camera used and the image evaluation system used, in addition, it would also be useful to include a description of the vacuum pump used (at least the type and parameters).
I think that in Table 3 it would be advisable to use specific values instead of numerical markings of individual variants in the case of markings at P, D and V. An example for P instead of 1, 2, 3, stop at 0.6, 0.75, 0.9 kPa and for D and V respectively. I believe that such a solution will increase the readability of the presented results. In this case, consider the necessity to present a 2nd vacuum table (at least type and parameters).
In line 336-337 it is stated that this applies to all cases, I believe it is better to apply the statement that this applies to most of the cases observed. A similar remark often applies to subsequent statements.
It would be useful to include information on the number and type of other particles sucked in, if any, at the vacuum set. Such particles will probably be easy to separate in, for example, the sieve separation process, but I think that such information could be important and interesting to evaluate the device in the presented work.
If such analyzes were conducted, I believe that they should be presented.I mean information about the damage to the tea leaves resulting from the applied vacuum harvesting. If such measurements were not carried out, I think that they should be in order to conduct more accurately analysis ofdevice's usefulness and the accuracy of its positioning.
I suggest the authors to consider including the results of multivariate analysis of variance. This information may facilitate the analysis of the obtained results and their correct interpretation.
Greetings
Reviewer 2 Report
Few general observations:
- It is recommended to express the values in SI units (ex. D=34 mm can be changed in D=0.034 m, etc.);
- Please, justify more about the option for Aduino 2560;
- If it is possible, at discussions, please give some information about the tea bud damages during harvesting (it could be useful for readers), also a comparison with other values regarding tea buds floating speed.
